# Osteoblast Dysfunction in Non-Hereditary Sclerosing Bone Diseases

**DOI:** 10.3390/ijms22157980

**Published:** 2021-07-26

**Authors:** Liberato Giardullo, Alberto Altomare, Cinzia Rotondo, Addolorata Corrado, Francesco Paolo Cantatore

**Affiliations:** Rheumatology Clinic, Department of Medical and Surgical Sciences, University of Foggia, 71122 Foggia, Italy; alto.albe@gmail.com (A.A.); cinzia.rotondo@gmail.com (C.R.); ada.corrado@unifg.it (A.C.); francescopaolo.cantatore@unifg.it (F.P.C.)

**Keywords:** bone sclerosis, melorheostosis, bone metabolism, osteoblasts, osteocondensation

## Abstract

A review of the available literature was performed in order to summarize the existing evidence between osteoblast dysfunction and clinical features in non-hereditary sclerosing bone diseases. It has been known that proliferation and migration of osteoblasts are concerted by soluble factors such as fibroblast growth factor (FGF), platelet-derived growth factor (PDGF), transforming growth factor (TGF), bone morphogenetic protein (BMP) but also by signal transduction cascades such as Wnt signaling pathway. Protein kinases play also a leading role in triggering the activation of osteoblasts in this group of diseases. Post-zygotic changes in mitogen-activated protein kinase (MAPK) have been shown to be associated with sporadic cases of Melorheostosis. Serum levels of FGF and PDGF have been shown to be increased in myelofibrosis, although studies focusing on Sphingosine-1-phosphate receptor was shown to be strongly expressed in Paget disease of the bone, which may partially explain the osteoblastic hyperactivity during this condition. Pathophysiological mechanisms of osteoblasts in osteoblastic metastases have been studied much more thoroughly than in rare sclerosing syndromes: striking cellular mechanisms such as osteomimicry or complex intercellular signaling alterations have been described. Further research is needed to describe pathological mechanisms by which rare sclerosing non hereditary diseases lead to osteoblast dysfunction.

## 1. Introduction

Sclerosing bone diseases are a heterogeneous group of skeletal alterations that share the process of impaired bone ossification. They are mostly rare diseases that are usually classified in hereditary and non-hereditary bone diseases. Hereditary sclerosing bone diseases are caused by genetic alterations resulting in increased bone formation and are represented by osteopetrosis, osteopoikilosis, pyknodysostosis, progressive diaphyseal dysplasia, osteopathia striata, hereditary multiple diaphyseal sclerosis andhyperostosis corticalis generalisata. Non-hereditary sclerosing bone diseases include intramedullary osteosclerosis, melorheostosis, Paget disease of bone, Erdheim-Chester disease, myelofibrosis and sickle cell disease. The diagnostic process of these conditions is often challenging. The resulting osteocondensation seen in imaging may be the final stage of an impaired process of stem cells differentiation which is in turn the consequence of altered levels of soluble growth factors. The basic laboratory tests in the diagnostic process of osteocondensation should include blood count, erythrocyte sedimentation rate, protid electrophoresis, transaminases, serum creatinine, calcium, bone alkaline phosphatases (bALP) and prostate specific antigen (PSA). Histology may also be required in order to exclude the diagnosis of hematological neoplasms. Impaired levels of soluble factors influencing osteoblast (Ob) metabolism such as fibroblast growth factor (FGF), platelet-derived growth factor (PDGF) or bone morphogenetic protein (BMP) or alterations of intracellular pathways such as Wnt signaling, Mitogen-activated protein kinase (MAPK) signaling, Receptor activator of NF-κB (RANK)/ RANK ligand (RANKL)/ Osteoprotegerin (OPG) pathway have been described in this group of diseases. Ob metastases showed peculiar mechanisms inducing osteosclerosis such as cross-talk between cancer cells and Ob. The aim of this review is to describe the main clinical features in non-hereditary sclerosing bone diseases and to review the existing evidence concerning the Ob dysfunction involved in the physiopathological mechanism of these disorders.

## 2. Osteoblast Physiology

Osteoblasts (Ob) originate from mesenchymal cells, secrete matrix proteins and promote mineralization during the bone modelling and restructuring process [1]. Ob are unable to function as a single cell, in fact they function in a group of cells and the functional unit made up of Ob and the bone produced is called bone multicellular units (BMU). The mineralized skeleton is the support for human body and is a fundamental store of calcium, phosphate, participating also to the basic-acid homeostasis [2]. Osteoclasts are the counterpart of Ob, being mainly involved in the bone resorption process. RANK is expressed on the surface of osteoclasts while RANKL, a trans-membrane protein produced in stromal cells and Ob, enhances the activation and differentiation of osteoclast by binding to RANK. OPG is secreted by Ob and stromal stem cells and prevents excessive bone resorption from skeleton by binding to RANKL and preventing the interaction with RANK [3]. Although RANK/RANKL pathway is mostly considered to be a key factor in the activation of osteoclasts, Sugamori et al. [4] demonstrated that RANKL-binding peptides WP9QY and OP3-4 may stimulate Ob proliferation and vesicular RANK secreted by the maturing osteoclasts; binding osteoblastic RANKL, they promote bone formation by triggering RANKL reverse signaling [5]. Ob proliferation and migration is influenced by soluble factors such as FGF, PDGF, BMP, parathyroid hormone (PTH), insulin-like growth factor (IGF), and the Wnt family of proteins. FGF stimulates Ob proliferation [6] and leads to a decreased apoptosis [7]; it has been known that PDGF increases both migration and proliferation in Ob [8]. MAPK or MAP kinase is a protein kinase that plays a leading role in the regulation of complex cell functions such as differentiation, proliferation or apoptosis [9]. Extracellular signal regulated kinase 1 and 2 (ERK 1/2) are members of MAPKs, which are activated by a kinase called MAPK/ERK kinase (MEK) in response to growth stimuli. It has been shown that FGF and PDGF activate ERK and consequently define Ob proliferation [10], although the precise mechanisms of this activation are not known. Kyono et al. [11] displayed that FGF2 upregulates genes involved in osteocyte differentiation in a MAPK dependent manner. Wnt proteins bind to surface receptors on mesenchymal cells such as Frizzled and LRP5, triggering the activation and nuclear translocation of the transcription factor β-catenin, which in turn defines the transcription of genes involved in Ob differentiation [12]. Dysregulation of these bone homeostasis control may be involved in the pathogenesis of bone sclerosing diseases. Constitutive activation of β-catenin has been linked to the pathogenesis of osteopetrosis, a disease included among hereditary sclerosing bone dysplasias [13].

## 3. Melorheostosis

Melorheostosis is a rare mesenchymal dysplasia consisting of a local massive sclerosis of the bone. Clinical findings of this disease depend on the location of the bone involved and include local pain, stiffness, limited range of motion, limb swelling. Long bones of the upper or lower extremities are usually involved, while axial skeleton is rarely affected [14]. Typically, melorheostosis appears by late childhood or adolescence [15] and is often segmental and unilateral, following a linear and segmental distribution of sclerotomes. No changes are expected in routine laboratory testing, although serum bALP may be increased. This disease shows peculiar radiographic findings. Irregular cortical sclerosis defines the “dripping candle wax sign” that differentiates it from osteoma, striated osteopathy, myositis ossificans and paraosteal osteosarcoma. Melorheostosis is a benign disorder and symptomatic treatment is usually sufficient. Psychosocial problems, limb deformities, nerve or tendon impingement may require surgery with decompression and debulking [16]. Somatic mosaicisms for MAP2K1-activating mutations are involved in the pathogenesis of melorheostosis; in fact, it has been shown that SMAD3 *p.S264* substitutions increase transforming growth factor β (TGF-β) signaling, which in turn leads to an increased Ob proliferation and a consequent bone sclerosis [17,18]. Post-zygotic mutations of MAP2K1 encoding MEK1-ERK1/2 proteins, are observed in most cases of sporadic melorheostosis. Immunohistochemical analysis of bone tissue from patients with MAP2K1 mutations showed an increased ERK1/2 expression in osteocytes and Ob compared to healthy controls [19] (Table 1 and Figure 1).

## 4. Intramedullary Osteosclerosis

Intramedullary osteosclerosis (IMOS) is an uncommon condition not associated with family history, systemic disease, or triggers. IMOS shows adult onset, female predominance and local pain; nevertheless, no specific signs and symptoms characterize this disease and the diagnosis can only be made after excluding other diseases. The lower limbs are usually involved and tibial diaphysis is the most commonly involved site [20]. The radiographic findings in IMOS consist of the formation of new endosteal bone and sclerosis in the diaphysis of long bones without soft tissue abnormalities; differential diagnosis includes osteoid osteoma, sarcoma, lymphoma, osteoblastic metastasis and healing stress fractures. The treatment strategy in IMOS has not yet been established. Conflicting data emerge from the use of non-steroidal anti-inflammatory drugs (NSAID) and greater relief is usually provided by surgery [35]. Biochemical analysis shows no alteration in bone metabolism. Histologic analysis shows non-specific changes in the bone tissue. The normal spongiosa is replaced by sclerotic, thickened trabeculae with a variable degree of mineralization; these aspects resemble those of melorheostosis. Bone scintigraphy can depict IMOS lesions even if asymptomatic [20]. Currently, no alteration of Ob metabolism or Ob activity has been demonstrated in IMOS.

## 5. Osteoblastic Metastasis

Prostate cancer (PCa) is the prototype of cancer with a predilection for generating osteoblastic metastases. The disordered bone architecture of bone metastases defines a structural weakness prone to fracture [21]. Since patients with metastatic bone prostate cancer have a relatively long survival [36], skeletal complications such as bone pain, hypercalcemia, impaired mobility, spinal cord compression may occur [37]. The genesis of osteoblastic metastases is a complex cascade of events involving both Ob and metastatic cells. A factor that may play a role in the development of osteoblastic PCa metastases is MDA-BF-1, a 45-kDa secreted form of the growth factor receptor ErbB3 expressed in PCa cells from metastatic bone lesions, but not in liver or lung metastases or in localized PCa cells [38]. Further evidence is needed to confirm a cause-and-effect relationship between the development of osteogenic metastases and the serum level of this factor. In vivo studies have shown that paracrine BMP signaling-mediated osteogenesis supports PCa progression in bone, since inhibition of the paracrine BMP4 decreases PCa tumor growth [22]. There is growing evidence that impaired osteoblastic function may be due to interaction with PCa, which secrete osteoblastic factors and induce Ob proliferation. PCa-118b is a patient-derived xenograft (PDX) generated from osteoblastic bone lesions [23], showing an increased expression of soluble factors belonging to the BMP/TGFβ and FGF family [24,39]. Wnt 7b is also highly expressed in PCa cells, thus promoting the development of osteoblastic lesions [40]. Growing evidence supports the role that endocellular vesicles (EV) may have in favoring the proliferation of PCa. PCa derived EV trains Ob towards a pro-tumorigenic cell type, creating a favourable niche for tumor growth [41]. Osteomimicry is a striking strategy adopted by PCa; in fact, they are able to express bone matrix proteins and interfere with the crosstalk of the bone cells [42], leading to an increased survival and proliferation of tumor cells. In vitro studies have shown that microRNA-141-3p contained in the exosomes from MDA PCa cells 2b is transferred to the Ob, promoting their activity and increasing OPG expression. Ob activity is significantly affected by changes in miR-141-3p levels. In vivo studies confirmed the important role played by microRNA in the genesis of bone metastases. In fact, intravenous injection of miR-141.3p mimic exosomes in mice develops apparent osteoblastic bone metastases at 4 weeks after injection [43]. The expression of VEGF by C4-2B PCa cells may also contribute to Ob proliferation in metastases, by linking VEGF receptor on the surface of Ob [44]. After a diagnosis of bone metastases, treatment is designed as a palliative and aims to reduce symptoms related to bone diseases such as fractures, pain or hypercalcemia [45]. Bisphosphonates are chemically stable derivatives of inorganic pyrophosphate which selectively bind the hydroxyapatite binding sites on bony surfaces, especially on surfaces undergoing active resorption. Then they are internalized in osteoclasts leading to the disruption of bone resorption [25]. Intravenous infusion of pamidronate has been shown to relieve skeletal pain in both lytic and sclerotic bone metastases [26]. Evidence of the anti-cancer effect was observed in breast cancer cells [46] and a meta-analysis showed disease-free survival benefits and a 15% improvement in overall survival [47]. Denosumab is a human monoclonal antibody that inhibits the activity of RANKL and osteoclasts; is injected subcutaneously in metastatic bone disease every 4 weeks and shows similar side effects to bisphosphonates such as osteonecrosis of the jaw, nausea, diarrhea and weakness [48]. Canabozantinib is a multikinase inhibitor targeting VEGFR2, MET, KIT, and mutationally activated RET. This therapy showed improvements in disease-free survival in patients with advanced PCa [49].

## 6. Myelofibrosis

Along with essential thrombocytopenia and polycythemia vera, idiopathic primary myelofibrosis (IMF) is a myeloproliferative neoplasm, a chronic disease of the bone marrow. Clinical manifestations in IMF include severe anemia, hepatosplenomegaly, bone pain, fatigue, night sweats, fever, cachexia and splenic infarction. Its prognosis is unfavorable, burdened by leukemic progression that occurs in 20% of patients, but also by the consequences of cytopenias such as infections or bleedings and the average survival is about 5 to 7 years. Bone marrow morphology is critical for a correct diagnosis showing bone marrow sclerosis associated with JAK2, CALR or MPL mutations observed in 90% of patients [27]. In IMF, early multipotent stem cells showed differentiation to megakaryocytes with dramatic expansion associated with fibrosis and the expression of proliferation-related genes [50]. Targeted therapies do not cause significant improvement in prognosis. In fact, the JAK inhibitor drug ruxolitinib, although initially shown to provide rapid improvement in constitutional symptoms and splenomegaly, is associated with serious adverse events such as thrombocytopenia or clinically significant anemia [51], unlike other myeloproliferative neoplasms, the osteosclerosis is a main feature of IMF, as 30 to 70% of IMF patients displayed osteosclerosis in roentgenographic studies [52], although the etiopathogenesis is not yet known. It has been hypothesized that TGFβ1 plays an important role in the pathogenesis of IMF by driving bone mesenchymal stem cells towards Ob differentiation and towards increased production of collagen [53]. IMF monocytes are overactivated and show increased TGF-β and IL-1 production compared to healthy controls [29]. Furthermore, TGFβ1 levels are related to megakaryocytic activity [30]. Wang et al. [54] showed that OPG serum levels are significantly higher in patients with IMF, in contrast to other myeloproliferative diseases, which did not differ from healthy controls. Moreover, a correlation was observed between sclerosis detected on CT (computer tomography) scan and OPG serum levels. Since increased serum levels of cytokines such as TGFβ, FGF [55] and PDGF [56] have been observed in patients with IMF, a possible stimulating effect on OPG production has been hypothesized [57].

## 7. Sickle Cell Disease

Sickle cell disease (SCD) is a group of blood inherited disorders characterized by the presence of at least one hemoglobin S allele (HbS) *p.Glu6Val* in HBB (Hemoglobin subunit Beta) and a second HBB pathogenic variant which results in the anomalous hemoglobin polymerization, thereby causing multiorgan infarctions. The most common type is known as sickle cell anemia. SCD encompasses several signs and symptoms such as intermittent vaso-occlusive events, chronic hemolytic anemia; bone involvement is the most common clinical manifestation [58], which can consist of both osteoporosis and osteosclerosis. Veskaridou et al. [59] assessed bone metabolism markers in a group of 52 patients with SCD and changes in bone mineral density (BMD) and found that 57.6% of patients had osteosclerosis (defined as a T-score of above +1 in either L1–L4 or the femoral neck) while 32.6% had osteopenia or osteoporosis (defined as T-score of below −1 in either L1–L4 or the femoral neck). Regardless of their BMD status, all SCD patients showed higher levels of serum OPG, bALP, C-terminal propeptide of collagen type-I, erythropoietin and serum soluble transferrin receptor compared to healthy controls. Interestingly, in patients with osteosclerosis, bone resorption as assessed by beta-C-terminal telopeptide levels was diminished. Bone ischemia after vaso-occlusive crises may be the mechanism involved in diminished bone resorption and consequently in the development of osteosclerosis, while osteoporosis could reflect bone loss resulting from bone marrow expansion. Mohktar et al. [60] demonstrated that iron overload inhibits Ob activity in vitro, resulting in potential osteoporosis. The infiltration of senescent neutrophils into the bone marrow of patients with SCD has been shown to be associated with a low number of Ob [61]. Hydroxyurea is an inductor of fetal Hemoglobin (HbF, α2γ2) and has been shown to be safe and efficacious treatment for most patients with SCD [62]. Novel treatments in SCD aim to inhibit HbS polymerization rather than increasing HbF. Voxelotor increases the oxygen affinity of the hemoglobin molecule and was recently approved in November 2019 [63].

## 8. Osteoarthritis

Osteoarthritis (OA) is a joint chronic disease, which causes damage to the cartilage and surrounding tissues. The pathogenesis of cartilage degradation is strictly related to concomitant changes in subchondral and periarticular bone, characterized by joint surface eburnation, subchondral irregular thickness, sclerosis, and osteophyte formation [64].

The main clinical symptoms are pain, stiffness and loss of function; radiological findings include joint space narrowing, geodes and subchondral bone sclerosis. Although risk factors such as high BMI, female gender, muscle weakness, genetic predisposition, prior joint injury are clearly associated with the development of OA [65], the exact etiopathogenic mechanisms of osteoarthritis are still not fully understood, but several studies have shown a central role played by Ob in the pathogenesis of bone changes in OA [66]. Genes involved in collagen-containing extracellular matrix, collagen binding, N-acetyglucosamine metabolic process, collagen fibril organization, skeletal system development, ossification, and Ob differentiation are upregulated [67]. Altered regulation of JUN, an important transcription factor, has been shown to play a crucial role in altered gene expression in OA. Periostin (PO), also called osteoblast-specific factor 2, has been shown to be upregulated in OA patients [32]. PO is also implicated in the activation of Wnt/beta catenin-signaling. PO-induced MMP-13 expression was inhibited by hydrobromide, an inhibitor of Wnt signaling [68]. A catabolic role for PO has therefore been suggested and the possibility of targeting this protein for protection against osteoarthritis has also been hypothesized. Mice lacking the PO gene showed significantly less cartilage damage than wild type [69].

It is known that an inflammatory process is associated with the alteration of the joint cartilage. OA is characterized by several stages; in the advanced stages, sclerosis of the subchondral bone is observed while in the early stages the subchondral bone thins [70,71]. In the advanced stages of OA, chondrocytes may influence Ob by increasing the expression of MMP-1 (matrix metalloproteinases); conversely, Ob may induce chondrocytes to increase MMP-13 expression and decrease aggrecan production [72]. Ob plays a major role in the development of osteo-proliferative changes in the bone such as Bouchard’s or Heberden’s nodules. The subchondral bone marrow spaces undergo major histopathological changes consisting of de novo bone formation and abundant infiltration of macrophages [73]. OA Ob has been shown to exhibit abnormal phenotypic features, such as strong expression of IGF-1 and impaired expression of RANKL and OPG [74], which may play a role in the genesis of bone sclerosis; moreover, according to a low or high endogenous production of prostaglandin E2 (PGE2) by OA Ob, two subgroups of OA patients can be identified. Low PGE2 Ob displayed lower OPG expression and increased RANKL level, conversely, high PGE2 Ob displayed higher OPG expression and decreased RANKL expression [33,66]. Osteotropic factors such as TNF-a, PGE2, IL-6, vitamin D-3 can induce an increased expression of RANKL [28,34]; their different expression of receptors and protein could explain the several types of lesions observed in OA. Treatment in osteoarthritis is mainly conservative: guidelines recommend NSAID as the first-line treatment and paracetamol as the second-line. Glucosamine did not differ from placebo for pain and its effect on structural progression of osteoarthritis is uncertain. In severe osteoarthritis, joint replacement should be considered [31].

## 9. Erdheim-Chester Disease

Erdheim-Chester Disease (ECD) is a rare form of non-Langerhans cell histiocytosis that was first reported by Chester and Erdheim in 1930 [75]. ECD usually shows multifocal xantho-granulomatous infiltration and almost every organ can be affected. Long bones are involved in 95% of patients [76] and on biopsy show foamy-to-epitheliod histiocytes surrounded by fibrosis [77]; exophthalmos occurs in some patients and is usually bilateral, symmetric and painless, and in most cases it occurs several years before the diagnosis, ‘coated aorta’ (circumferential soft-tissue sheathing of the thoracic aorta) and ‘hairy kidney’ (soft tissue rind of perirenal infiltration) are typical features in patients with ECD [78]. Both features could be a display of a retro-mediastinal and retro-peritoneal localization of the cancer and occurs in one third of the patients [76]. Cortical osteosclerosis is detected on plan radiographs, CT, MRI (magnetic resonance imaging) and on nuclear medicine techniques. Estrada-Veras et al. [76] found that technetium-99 bone scan was the most sensitive imaging technique in detecting bone disease compared to standard bone plain x-ray. Gastrointestinal system involvement is uncommon, showing lamina propria histiocyte infiltration on gastric biopsies [79]. Diagnosis of ECD requires typing of histiocytes in tissues: these are typically foamy and CD68+ CD1a− in ECD, whereas in Langerhans cell histiocytosis (LCH), they are CD68+ CD1a+ [80]. The advent of molecular medicine with targeted therapies has been defined in the last 5 years as a completely new approach in the treatment of this rare histiocytosis. In fact, according to genetic mutations shown by the patients, inhibitors of the MAPK pathway and the BRAF pathway are actually the first-line treatment, as the most relevant activating mutation is *BRAFV600E*, shown in 57-75% of cases [81]. This mutation is associated with uncontrolled cell growth via the MAPK-ERK pathway. In *BRAF-V600*-mutant ECD Patients with severe heart or neurological involvement, BRAF-inhibitors such as vemurafenib or dabrafenib are recommended [82]; if this mutation is absent, new generation sequencing techniques can be used to identify alterations of the MAPK pathway and support treatment with MAPK inhibitors (trametinib, cobimetinib, binimetinib) [83].

## 10. Paget Disease of the Bone

Paget disease of the bone (PDB) is a metabolic bone disorder characterized by an unbalanced turnover of bone tissue affecting one or more bones. Bone pain is the most common symptom and typically worsens at rest or during the night. In a series of cases, deafness was evaluated in 7.9% of the patients [84]. Entrapment of the cranial nerves in their foramina, a characteristic facial deformity known as “leontiasis ossea” and the involvement of the jawbones can also be observed. The bALP levels are increased in these patients, reflecting disease activity and being useful for follow-up, while the serum calcium titer is usually normal. The differential diagnosis includes bone metastases, osteopetrosis and McCune Albright syndrome. First line therapy of PDB is mainly supportive and involves the use of bisphosphonates, in order to inhibit osteoclast activity.

Focal bone changes are characterized by hypertrophic and giant osteoclast clusters containing up to 100 nuclei, while normal osteoclasts show up to 20 nuclei [85]. Ob hyperactivity is considered to be secondary to osteoclasts overactivation. A possible link between these two bone cell subtypes could be identified in sphingosine-1-phosphate (S1P), a sphingolipid involved in the production of RANKL in the Ob and in Ob differentiation [86]. Nagata et al. [87] showed higher S1P expression in osteoclasts of patients with PDB compared with normal donors. This sphingolipid defines an overexpression of the S1P receptor-3 on the surface of Ob in patients with PDB, which was higher than in healthy controls. Ob do not change morphologically, but their accelerated activity induces the formation of a disordered and woven lamellar bone resulting in weak tissue with a higher risk of stress fracture. A role played by the measles virus (MV) in PDB etiopathogenesis has been speculated since the 1970s, in fact sequences of MV messenger RNA (mRNA) have been observed in up to 90% of osteoclasts and other mononuclear cells in Pagetic bone samples. Pagetic osteoclasts expressing the MVNP (measle virus nucleocapsid protein) gene enhance the synthesis of soluble factors that induce osteoclast differentiation as well as the proliferation of Ob precursors and the constitutive expression of RANKL, which is a cornerstone in osteoclasts proliferation [88]. Bisphosphonates such as zoledronic acid or pamidronate represent the mainstay of the treatment, which is necessary in case of clinical manifestations such as pain, increased bALP serum levels or hypercalcemia, while asymptomatic patients do not require any treatment [89].

## 11. Conclusions

The pathogenic mechanisms underlying non hereditary sclerosing bone diseases are not yet fully understood. Several diseases in this group such as osteoarthritis or osteoblastic metastases have been better characterized thanks to more in-depth research, while the rare sclerosing bone syndromes need further research in order to enlighten the underlying alterations. Several intracellular proteins and specific gene expression in Ob have been shown to play a major role in the development of these syndromes. In melorheostosis, SMAD3 mutations may be the main factor in causing increased proliferation of Ob. Activation of the MAPK pathway may play a major role in the genesis of sclerotic lesions. Upregulation of MAPK was observed in both melorhestosis and ECD; in this hystiocytosis the BRAFV600E mutation is associated with a constitutive activation of MAPK signal. Wnt proteins play a leading role in regulating osteocyte proliferation and differentiation; dysfunctions of these proteins have been observed in the genesis of sclerosing lesions in osteoblastic metastases and osteoarthritis, where upregulation of PO defines the activation of wnt/beta-catenin signaling. Ob hyperactivity in PDB may result from increased expression of S1P on the surface of osteoclasts. In sickle cell anemia, iron overload has been shown to hamper osteoblastic activity. High serum levels of FGF, PDGF and TGFβ could drive the sclerosing process in bone marrow of patients with IMF, while OPG level was decreased. Further research is needed to understand the etiopathogenetic mechanisms of non-hereditary sclerosing bone diseases, as well as to develop targeted therapies to reduce bone complications and improve the outcomes of this set of diseases.

## Figures and Tables

**Figure 1 ijms-22-07980-f001:**
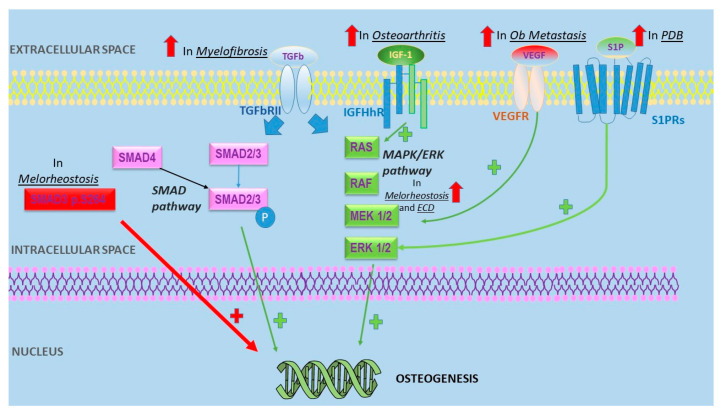
Schematic representation of several altered intracellular mechanisms in Ob underlying the etiopathogenesis of osteosclerosis in various non-hereditary sclerosing bone diseases. Abbreviations: PDB: Paget’s disease of the bone; ECD: Erdheim-Chester Disease; Ob: osteoblast.

**Table 1 ijms-22-07980-t001:** Pathogenic mechanisms, hypothesis or evidence defining osteosclerosis in non hereditary sclerosing bone diseases.

Disease	Pathogenic Mechanisms Leading to Osteosclerosis	References
Melorheostosis	-Mutations of MAP2K1 encoding MEK1-ERK1/2 proteins -SMAD3 *p.S264* substitutions were shown to increase TGF-β signaling	[16,17]
Osteoblastic metastases	-MDA-BF-1 expressed only in PCa bone metatasis -BMP paracrine effect on PCa -Cross-talk between PCa and osteoblasts -Osteomimicry -Micro-RNA transferring from PCa to osteoblasts -Altered VEGF expression	[19,20,21,22,23,24]
Myelofibrosis	-Correlation between bone sclerosis and OPG serum level -High FGF and TGF levels	[25,26]
Sickle cell disease	-Increased activity after osteoblast inhibition by iron overload -RANK/RANKL/OPG and Wnt/β-catenin pathways alterations	[27,28]
Osteoarthritis	-JUN disregulation -Periostin upregulation -IGF-1 expression may promote bone sclerosis	[29,30,31]
Erdheim Chester disease	Activation of MAPK pathway	[32]
Paget disease of the bone	-Sphingosine-1-phosphate (S1P) overexpression leads to osteoblasts hyperactivity -MV mRNA commonly observed in Pagetic osteoclasts	[33,34]

Abbreviations: MAP2K1: mitogen-activated protein kinase; MEK1-ERK: MAPK/ERK kinase–extracellular signal regulated kinase; BMP: bone morphogenetic protein; PCa: Prostatic Cancer; FGF: fibroblast growth factor. TGF: transforming growth factor; VEGF: Vascular endothelial growth factor; OPG: Osteoprotegerin; MV: measle virus.

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
