# Peer review of "Osteoblast Dysfunction in Non-Hereditary Sclerosing Bone Diseases"

_ijms, 2021, doi:10.3390/ijms22157980_

Round 1
Reviewer 1 Report
The review needs extensive editing of English language by a native speaker. There are multiple spelling errors throughout the manuscript that make the comprehension difficult. Some examples are reported below:
-The term evidence is singular not plural. Please correct it.
-Line 29-34: sentence too long
-Line 35: conditions
-Line 125-130: sentence too long
-Please define osteoblast in the introduction and keep the abbreviation throughout the manuscript. For example, In the chapter 4 osteoblast is not defined.
-Please uniform the following terms: osteoblastic differentiation/proliferation or Ob differentiation/proliferation.
-The first part of the chapter 2 lacks appropriate references
The review is a simple description of non-hereditary sclerosing bone diseases. The review completely lacks the description of the clinical aspects of these diseases such as standard therapy and emerging treatments. Moreover, I suggest the authors to add one or two figures summarizing the main topic of the review as well as a brief overview of the pathogenesis of such diseases.
Lastly, the main mechanisms underlying osteoblast disfunction should be added in the first paragraphs.
In this form, the present review does not provide a significan contribution to the field.
Reviewer 2 Report
This review describe the main clinical features in non-hereditary sclerosing bone diseases. Authors also addressed the existing evidences concerning the osteoblast dysfunction involved in the physiopathological mechanism of these diseases, further detailing the proteomic and genetic pathways. It is a easy to follow review that provides a brief view of rare sclerosing non hereditary diseases, an issue that is not very familiar to a high percentage of health-related readers.
Author Response
Dear Reviewers,
We thank you for the generous comments on the manuscript: “Osteoblast dysfunction in non-hereditary sclerosing bone diseases”.
A revision of the spelling and grammar errors has been made. Moreover, we added lacking therapeutic and clinical aspects lacking in the first version of the manuscript. Moreover, a simplified figure explaining the common underlying pathogenic pathways of the diseases has been added.
Sincerely,
Dr. Liberato Giardullo
Round 2
Reviewer 1 Report
The authors answered all my comments. But I can't find the figure
Author Response
Dear Reviewer,
Attached you can find a word document containing the figure.
An error occurred during the upload of the figure, now it is also in the main document.
Kind Regards

Round 3
Reviewer 1 Report
Agree